# Relative Leukocyte Telomere Length and Telomerase Complex Regulatory Markers Association with Leber’s Hereditary Optic Neuropathy

**DOI:** 10.3390/medicina58091240

**Published:** 2022-09-07

**Authors:** Rasa Liutkeviciene, Rasa Mikalauskaite, Greta Gedvilaite, Brigita Glebauskiene, Loresa Kriauciuniene, Reda Žemaitienė

**Affiliations:** 1Department of Ophthalmology, Lithuanian University of Health Sciences, 44307 Kaunas, Lithuania; 2Neuroscience Institute, Lithuanian University of Health Sciences, 44307 Kaunas, Lithuania

**Keywords:** Leber’s hereditary optic neuropathy (LHON), relative leukocyte telomere length (RLTL), idebenone, telomerase complex

## Abstract

*Background and Objectives:* To evaluate the association of relative leukocyte telomere length (RLTL) and telomerase complex regulatory markers with Leber’s hereditary optic neuropathy (LHON). *Material and Methods:* A case-control study was performed in patients with LHON (≥18 years) and healthy subjects. The diagnosis of LHON was based on a genetic blood test (next-generation sequencing with Illumina MiSeq, computer analysis: BWA2.1 Illumina BaseSpace, Alamut, and mtDNA Variant analyzer 1000 were performed) and diagnostic criteria approved by the LHON disease protocol. Statistical analysis was performed using the standard statistical software package, IBM SPSS Statistics 27. Statistically significant results were considered when *p* < 0.05. *Results:* Significantly longer RLTL was observed in LHON patients than in healthy controls (*p* < 0.001). RLTL was significantly longer in women and men with LOHN than in healthy women and men in the control group (*p* < 0.001 and *p* = 0.003, respectively). In the elderly group (>32 years), RLTL was statistically significantly longer in LHON patients compared with healthy subjects (*p* < 0.001). The GG genotype of the *TERC* rs12696304 polymorphism was found to be statistically significantly higher in the LHON group (*p* = 0.041), and the C allele in the *TERC* rs12696304 polymorphism was found to be statistically significantly less common in the LHON group (*p* < 0.001). The RLTL of LHON patients was found to be statistically significantly longer in the *TERC* rs12696304 polymorphism in all tested genotypes (CC, *p* = 0.005; CG, *p* = 0.008; GG, *p* = 0.025), *TEP1* rs1760904 polymorphism in the GA genotype (*p* < 0.001), and *TEP1* gene rs1713418 in the AA and AG genotypes (*p* = 0.011 and *p* < 0.001, respectively). *Conclusions:* The RLTL in LHON patients was found to be longer than in healthy subjects regardless of treatment with idebenone. The *TERC* rs12696304 polymorphism, of all studied polymorphisms, was the most significantly associated with changes in LHON and telomere length.

## 1. Introduction

Leber’s hereditary optic neuropathy (LHON) is one of the most common diseases caused by mutations in mitochondrial deoxyribonucleic acid (mtDNA) [1,2]. Symptomatic LHON disease leads to progressive, irreversible loss of visual acuity, including total blindness, and dramatically impacts the quality of life of affected patients [3,4,5,6]. The first symptoms of the disease usually appear in young adult males in the second to third decade of life, and 90% of affected individuals lose their vision before the age of 50 [2,7,8,9,10]. LHON is caused by a genetic mutation in the mitochondrial genome. The most common mutations are m.11778G > A, m.3460G > A, and m.14484T > C. In LHON, mutations in mtDNA in the retinal ganglion cells (RGC) lead to a decrease in adenosine triphosphate (ATF) synthesis, increased oxidative stress, and degeneration and apoptosis of these cells [11]. The definitive diagnosis of LHON is made when ophthalmologic features characteristic of the disease appear and/or a pathogenic variant of mtDNA is detected by molecular genetic testing. Evaluation of visual acuity is the primary goal in the treatment of this disease, and to date, the only approved drug for the treatment of LHON is idebenone. Idebenone, an analogue of coenzyme Q10, helps to restore impaired ATF production in retinal cells and activate the function of still viable but inactive cells [12,13]. Patients treated with idebenone have noted improvement in vision, but blindness due to optic nerve cell death remains a pharmacological challenge [2,14].

Telomeres are nucleoprotein complexes at the ends of eukaryotic chromosomes. The length of telomeres has been associated with a variety of diseases and pathologies [15,16]. Evidence suggests that relative leukocyte telomere length (RLTL) is related to age, sex, various oncologic and chronic diseases, diet and lifestyle, changes in telomere lengths associated with mitochondrial dysfunction, and longer telomere lengths in optic nerve and inflammatory eye diseases (Birdshot uveitis) [17]. Repeated telomere sequences at the ends of chromosomes are synthesized by telomerase [18]. Regulatory proteins inhibit their activity. Therefore, polymorphisms in the *TERC* gene rs12696304, rs35073794, and *TEP1* gene rs1713418, rs1760904 are thought to affect telomerase activity and closely related to changes in RLTL [19,20,21].

In studying the pathogenesis of LHON, the most important changes occur in mitochondria, whose dysfunction can be associated with altered RLTL. The pathophysiology of the disease is complex, and selective damage to RGCs associated with LHON-specific mitochondrial dysfunction has not been fully elucidated [5]. Given the lack of studies examining the relationship between telomere length and damage to ocular structures, we decided to establish links between RLTL and LHON. It is important to continue the study of this disease at the molecular level, as the rapid development of research promotes the search for genetic markers that could potentially contribute to the discovery and development of treatments, and a better understanding of the pathogenesis mechanism would lead to earlier diagnosis.

## 2. Materials and Methods

### 2.1. Ethics Statement

All subjects have signed an agreement according to the Declaration of Helsinki (Nr-BE-2-102, approved date: 6 September 2019). The study was managed in the Department of Ophthalmology, Lithuanian University of Health Sciences. The control group included healthy people with no ocular pathology at an examination.

The exclusion criteria for Leber’s hereditary optic neuropathy (LHON) patients were:Under the age of 18;Systemic diseases (e.g., malignant tumors, systemic connective tissue diseases, diabetes mellitus, chronic infections, or conditions following organ or tissue transplantation).

### 2.2. Leber’s Hereditary Optic Neuropathy (LHON) Diagnostics

Individuals with LHON mutations are generally divided into asymptomatic mutation carriers and symptomatic patients with ocular lesions. The reason some carriers never experience symptoms is still unknown, but genetic factors and environmental influences, alcohol, and smoking are thought to play an important role [22].

The classic clinical manifestation of LHON is subacute, painless loss of central vision in one eye followed by loss of vision in the other eye over days, weeks, or months. It is essential to pay attention to macular and papillomacular retinal ganglion cells (RGC) due to the degeneration of these cells by optical coherence tomography (OCT). The common findings in asymptomatic mutation carriers and early subacute stage are optic nerve disc (OND) telangiectasia microangiopathy and OND swelling (pseudoedema) [5,23,24]. As the disease progresses, rapid axonal loss occurs first, followed by diffuse visual atrophy. The chronic stage of the disease is reached approximately 1 year after onset [5,23,24,25].

The final diagnosis of LHON is made when the probe exhibits ophthalmological features characteristic of the disease and/or one of the pathogenic variants of mtDNA is detected by molecular genetic testing. 

A neuro-ophthalmology examination and next-generation sequencing (NGS) with Illumina MiSeq, computer analysis: BWA2.1 Illumina BaseSpace, Alamut, mtDNA Variant analyser 1000 were performed. Tested genes: *MT-ND1*, *MT-ND2*, *MT-ND3*, *MT-ND4*, *MT-ND4L*, *MT-NDS*, *MT-ND6*, *MT-CYB*, *MT-CO1*, *MT-CO2*, *MT-CO3*, *MT-ATP6*, *MT-ATP8*, *MT-RNRI*, *MT-RNR2*, *MT-TA*, *MT-TT*, *MT-TP*, *MT-TE*, *MT-TL2*, *MT-TS2*, *MT-TH*, *MT-TR*, *MT-TG*, *MT-TK*, *MT-TD*, *MT-TS1*, *MT-TC*, *MT-TY*, *MT-TN*, *MT-TW*, *MT-TM*, *MT-TI*, *MT-TQ*, *MT-TL1*, *MT-TF*, *MT-TV* for all the patients.

### 2.3. DNA Extraction, RLTL Measurement, and Genotyping

DNA extraction, relative telomere leukocyte length measurement, and analysis of *TEP1* rs1760904, rs1713418, *TERC* rs12696304, and rs35073794 were carried out at the Laboratory of Ophthalmology, Neuroscience Institute, Lithuanian University of Health Sciences. DNA was extracted using the DNA salting-out method from venous blood samples.

SNPs were determined using TaqMan^®^ genotyping assays (Thermo Scientific, Pleasanton, CA, USA). RLTL measurement and *TEP1* rs1760904, rs1713418, *TERC* rs12696304, and rs35073794 genotyping were conducted using real-time PCR according to the manufacturer’s recommendations using a Step One Plus real-time PCR system (Applied Biosystems, Foster City, CA, USA).

### 2.4. Statistical Analysis

Statistical analysis was conducted with SPSS/W 27.0 software (Statistical Package for the Social Sciences for Windows, Inc., Chicago, IL, USA). Gender distribution was presented as absolute numbers with percentages and compared with the χ^2^ test. Data, which are not ordinally distributed between the two groups or subgroups, were measured with the Mann-Whitney U test.

We conducted a Hardy-Weinberg analysis to analyze the observed and expected frequencies of *TEP1* rs1760904, rs1713418, *TERC* rs12696304, and rs35073794 with the χ^2^ test in the control group. The frequencies of *TEP1* rs1760904, rs1713418, *TERC* rs12696304, and rs35073794 between groups were analyzed with the χ^2^ test. In addition, binary logistic regression analysis was performed to assess the influence of genotypes on the development of PA. The odds ratios (OR) and 95% confidence intervals (CI) are reported. The Akaike Information Criterion (AIC) selected the best genetic model. Statistically significant differences were considered when *p* < 0.05. We also presented an altered significance threshold for multiple comparisons, alpha = 0.017 (0.05/3) since we assessed three SNPs after excluding *TERC* rs35073794.

## 3. Results

The study included 144 subjects who were divided into two groups: a control group (n = 130) and a group of patients with Leber’s hereditary optic neuropathy (LHON) (n = 14). Genotyping of *TEP1* rs1760904, rs1713418, *TERC* rs12696304, and rs35073794 polymorphisms was performed after the formation of the study groups. The LHON group of patients consisted of 14 subjects: 10 men (71.4%) and 4 women (28.6%). The mean age of the patients was 40.6 years. The control group consisted of 130 subjects: 61 men (46.9%) and 69 women (53.1%). The average age of the control group was 39.9 years. The study groups did not differ when compared by gender and age (*p* = 0.081, *p* = 0.843, respectively). The relative leukocyte telomeres length (RLTL) was determined in the subjects, and a comparison showed that the RLTL was statistically significantly longer in patients with LHON than in the control group (*p* < 0.001). The demographics of the subjects are presented in Table 1.

After estimating the frequencies of the genotypes according to the Hardy-Weinberg equilibrium (HWE), the results showed that *TERC* rs35073794 deviated from HWE (*p* < 0.05) (Table 2). This distribution of genotypes and alleles is possible due to the small sample size, so the sample should be increased in future studies [26]. Therefore, SNP was not included in the further analysis.

### 3.1. Influence of TEP1 rs1760904, rs1713418, TERC rs12696304 on LHON Development

The genotype and allele analysis revealed that the *TERC* rs12696304 GG genotype was more frequent in the LHON group than in the control group (21.4% vs. 6.2%, *p* = 0.041). In addition, the C allele was found to be statistically significantly less frequent in the LHON group compared with the control group (36.0% vs. 73.1%, *p* < 0.001) (Table 3).

### 3.2. Differences in Relative Leukocyte Telomere Length (RLTL) between LHON and Healthy Subjects

Analyzing the relative length of leukocyte telomere length, we found statistically significant differences between LHON and the control groups (median (IQR): 4.123 (9.039) vs. 0.613 (0.582), *p* < 0.001). The results are shown in Figure 1.

A comparison of RLTLs between different genotypes for all polymorphisms studied was also performed. RLTL was found to be statistically significantly longer in LHON than in controls with the *TEP1* rs1760904 GA genotype (median (IQR): 4.505 (15.405) vs. 0.696 (0.549), *p* < 0.001). It was also statistically significantly longer in LHON than in controls with *TEP1* rs1713418 AA (median (IQR): 3.474 (16.635) vs. 0.506 (0.682), *p* = 0.011) and AG genotypes (median (IQR): 7.940 (12.964) vs. 0.644 (0.565), *p* < 0.001). RLTL was also found to be statistically significantly longer in LHON than in controls with *TERC* rs12696304 CC (median (IQR): 3.097 (3.780) vs. 0.456 (0.652), *p* = 0.005), CG (median (IQR): 12.105 (16.187) vs. 0.694 (0.573), *p* = 0.008) and GG genotypes (median (IQR): 9.764 (-) vs. 0.982 (1.529), *p* = 0.025) (Table 4).

## 4. Discussion

In Lithuania, LHON disease has been diagnosed in 19 individuals since 2018, of whom 17 patients (13 men and 4 women) are monitored and treated at the Department of Ophthalmology, Lithuanian University of Health Sciences. LHON patients’ ages in Lithuania range from 9 to 61 years depending on the onset time (our study includes patients aged 19 to 61 years), and the literature describes cases in which the age varies from 2 to 87 years [27,28]. The mean age of LHON patients in our study was 38.93 years (SD = 14.69), but we included only adult patients older than 18 years, and the mean age of all patients currently treated in Kaunas clinics was 21.64 years. In the literature, an age of 22–25 years has been reported for the onset of LHON [27,28].

The male-to-female ratio was 2.75:1 in our study and 3.25:1 for all patients monitored at the eye clinic LUHS KC, with the majority of patients being male. These results are not significantly different from studies in Denmark (3.72:1) and the northwest of England (3.32:1) [27]. In our LHON study group, the proportion of males was 73.3 percent (76.5% of all patients treated for LUHS in that year), and the research data indicate that approximately 90% of patients are men [8]. This may be due to the fact that in Lithuania, LHON has only been diagnosed since 2018 and not all patients with LHON have been identified yet.

According to the data of this study, the diversity and prevalence of mutations in our country are slightly different from the three most common mtDNA mutations reported in the literature. The literature emphasizes that more than 90% of all patients with LHON (about 70% in Northern Europe) have one of the following three pathogenic variants: m.11778G > A (*MT-ND4*), m.14484T > C (*MT-ND6*), or m.3460G > A (*MT-ND1*) [1,6,24,29,30,31,32,33,34], but in our study in Lithuania, in patients with LHON > 18 years, these main mutations were detected only in 33.4%. (41.2% of all patients treated in LUHS KC in that year). From the available data, the most common LHON-causing mutation, m.11778G > A (*MT-ND4*), was diagnosed in 5 out of 17 patients (29.4%) in LUHS, which is in contrast to global statistics, in which this mutation is found in about 60% of diseased Northern Europeans and 90% of diseased Asians [7,30,31,34]. It is important to note that a relatively large variety of other (non-primary) mtDNA mutations leading to LHON disease were observed during the study survey. As many as four-tenths (40%) of the mutations identified in the study have not yet been included in the official list of pathogenic mutations in MitoMap, but have been described in studies as mutations associated with LHON (https://www.mitomap.org (accessed on 16 January 2022)). The following mutations are not included in the list: m.3394T > C (*MT-ND1*), m.3866T > C (*MT-ND1*), m.8836A > G (*MT-ND4*), and an autosomal recessive mutation in the *DNAJC30* gene that was first described in the literature only in 2021 [35].

One of the main tasks of our study was to link RLTL to LHON. Prior to this study, shorter RLTL lengths were expected to be found in patients with primary mitochondrial disorders (e.g., diseases caused by mutations in the mitochondrial genome) and shorter telomeres compared with the healthy population, as described in the literature [36]. The literature also mentions that telomeres are sensitive to oxidative damage, leading to an increase in the rate of telomere shortening [37,38]. A study by Rovcanin et al. found that LHON patients have an increase in total oxidant status (TOS) and a decrease in total antioxidant status (TAS) levels, which suggests the existence of substantial oxidative stress [39]. As both telomeres and LHON patients are sensitive to oxidative stress, it could be an essential factor in the pathogenesis of the disease. One of the mechanisms in the pathogenesis of LHON is electron transport chain (ETC) dysfunction caused by primary mtDNA mutations, resulting in decreased ATF production, excessive secretion of ROS, and death of RGC [40]. However, this theory could not be confirmed in our study, and a statistically significant higher incidence of RLTL was observed in LHON patients than in healthy controls. This prompted a more detailed analysis of the literature, as no studies on the association between RLTL and LHON disease have been published to date. In the literature, an association between a genetically determined longer telomere length and an increased risk of certain oncological processes was observed [41], with lower stress levels, physical activity, a high-quality diet (e.g., with free ω-3 fatty acids, intake of some antioxidants, and low consumption of processed red meat), and adequate sleep associated with longer telomeres [42,43]. Studies of ocular structures showed that telomeres were longer in people with uveitis than in healthy controls (in cases of chronic inflammation) [17]. Long telomeres were observed in the neurosensory part of the retina (longer telomere lengths may be related to the fact that metabolic processes are most active in this part of the retina) [44]. In the pathogenesis of LHON, retinal ganglion cells are the most vulnerable, and therefore a longer RLTL was observed in patients with LHON, which stimulates discussion, and a larger sample of studies should be investigated in the future.

In our study, patients were categorized by sex and age, and the literature emphasizes the dependence of RLTL length on these two parameters. Previous studies showed that women’s telomeres are generally longer than those of men, and this phenomenon increases with age [45]. However, in our study, no statistically significant association was found between RLTL in relation to sex within the LHON or control groups; and statistically significantly longer RLTLs were found between LHON men and women compared with healthy control men and women, respectively.

The length of telomeres and their attrition rate were proposed in the literature as biological markers of age-related disease risk [46]; therefore, the data obtained in this study were disaggregated by age group. There was no statistically significant difference in RLTL between LHON and control subjects in the younger group (<32 years), but RLTL was statistically significantly longer in LHON patients compared with controls in the older group (>32 years). A study of ocular structures investigated whether chronological aging is associated with the shortening of telomere length in the human retina. However, no significant differences were found, suggesting that age does not affect telomere length in retinal neuronal cells [44]. Another important question in the study was whether treatment with idebenone, a synthetic coenzyme Q10 analogue, could affect RLTL. In the literature, idebenone was reported to act as an antioxidant and inhibit the formation of excess ROS, and the use of some antioxidants was associated with longer telomeres [41,42]. Therefore, in this study, we wanted to determine whether there was a statistically significant difference in the RLTL of LHON patients who received blood before treatment with idebenone compared with other LHON patients treated with idebenone, but no statistically significant difference was found. This suggests that a statistically longer duration of treatment with idebenone was observed in patients with RLTL compared with the controls.

To our knowledge, no studies have been performed to analyze the association of the *TEP1* rs1760904, rs1713418, and *TERC* rs12696304, rs35073794 gene polymorphisms with LHON. It should be noted that genetic variants affecting telomere length, telomerase activation, telomeric protein configuration, and telomere-related gene SNPs can cause functional alterations responsible for the risk of developing cancer and a wide range of other telomere and oxidative stress-related pathologies [47,48,49,50,51,52].

In our study, we found that *TERC* rs12696304 can influence the occurrence of LHON. The GG genotype of the polymorphism is statistically significantly higher in the LHON group compared to that of the control group (*p* = 0.041). In addition, the C allele of the *TERC* polymorphism rs12696304 was found to be statistically significantly lower in the LHON group than in the control group (*p* < 0.001). It is important to mention that in the study of RLTL in LHON of this polymorphism, statistically significantly longer telomeres were found in all studied genotypes compared to healthy controls (CC *p* = 0.005; CG *p* = 0.008; GG *p* = 0.025). This is consistent with the exceptional relationship between SNPs and telomere lengths reported in the literature for this particular polymorphism (*TERC* rs12696304) [53].

In our study, we also found an association between other polymorphisms and RLTL in individuals with LHON. The GA-genotype of the *TEP1* gene rs1760904 showed a statistically significant longer RLTL in LHON patients compared with controls (*p* < 0.001) and the AA- and AG-genotypes of the *TEP1* gene rs1713418 (*p* = 0.011 and *p* < 0.001, respectively).

The regulatory markers of the telomerase complex included in the study (*TEP1* rs1760904, rs1713418; *TERC* rs12696304) are useful to explain the factors leading to changes in RLTL, and the association of these polymorphisms with LHON has not been previously studied in the literature. The crucial changes in RLTL of these polymorphisms have been largely associated with tumor processes. It would be useful to conduct further scientific research studies in the context of various pathologies for which statistically significant changes in RLTL have been identified. Telomere length is measured in adults (>18 years) and provides only partial information about the dynamics of RLTL. Repeated lifetime measurements that take into account genetic factors and the effects of early childhood would provide a better understanding of telomere dynamics in adulthood, although such data collection and analysis would be very complex [54]. Blood samples were collected from our subjects at a specific age and different stages of LHON disease to measure RLTL. The literature emphasizes that telomere lengths are dynamic and vary due to a variety of factors as mentioned above, and patients with LHON should be the subject of further RLTL-related studies to monitor disease-related and treatment-related changes in RLTLs as a function of age, the onset of LHON, duration of treatment, and efficacy; and enroll as many subjects as possible in the study.

## Figures and Tables

**Figure 1 medicina-58-01240-f001:**
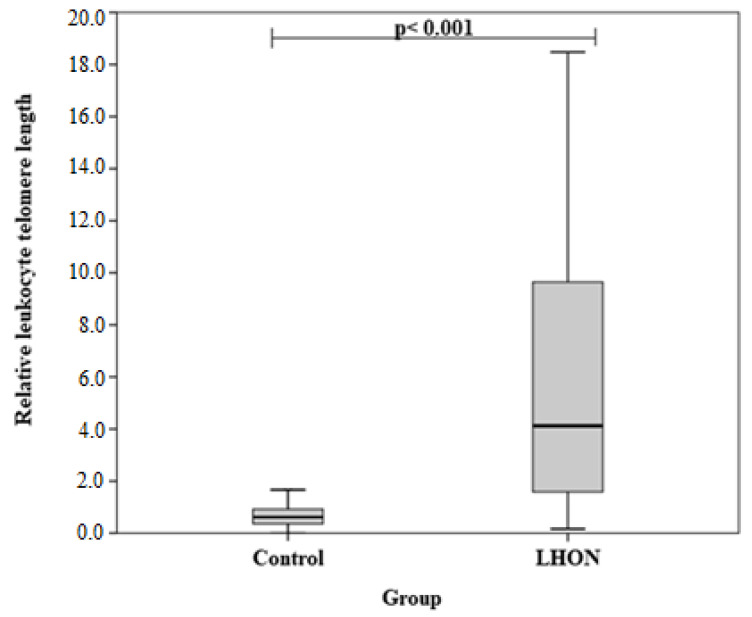
Relative length of leukocyte telomeres in LHON and control groups. The Mann-Whitney U test was used.

**Table 1 medicina-58-01240-t001:** Demographics.

Characteristics		Group		*p*-Value *
		LHON n (%)	Control n (%)	
Gender	Males	10 (71.4)	61 (46.9)	0.081
	Females	4 (28.6)	69 (53.1)	
Age, mean (IQR)		40.6 (13.1)	39.9 (17.9)	0.843
RLTL, mean (SD)		7.31 (7.28)	0.77 (0.75)	<0.001

* *p*-significance level and Bonferroni-corrected significance level when *p* = 0.05/3.

**Table 2 medicina-58-01240-t002:** Hardy-Weinberg equilibrium analysis.

Gene, SNP	Allele Frequencies		Genotype Distribution	*p*-Value *
*TEP1* rs1760904	0.49 (A)	0.51 (G)	30/69/31	0.482
*TEP1* rs1713418	0.43 (G)	0.57 (A)	21/71/38	0.204
*TERC* rs12696304	0.27 (G)	0.73 (C)	8/54/68	0.526
*TERC* rs35073794	0.19 (A)	0.81 (G)	0/49/81	<0.05

* *p*-significance level and Bonferroni-corrected significance level when *p* = 0.05/3.

**Table 3 medicina-58-01240-t003:** Influence of TEP1 rs1760904, rs1713418, TERC rs12696304 on LHON development.

Model	Genotype/Allele	OR (95% CI)	*p*-Value	AIC
*TEP1* rs1760904				
Codominant	AG vs. GG AA vs. GG	1.647 (0.429–6.323) 0.000 (0.000–)	0.467 0.998	88.358
Dominant	AG+AA vs. GG	1.148 (0.301–4.380)	0.840	93.812
Recessive	AA vs. GG+AG	0.000 (0.000–)	0.998	86.926
Overdominant	AG vs. AA+GG	3.242 (0.864–12.162)	0.081	90.283
Additive	A	0.622 (0.266–1.454)	0.274	92.626
*TEP1* rs1713418				
Codominant	AG vs. AA GG vs. AA	1.070 (0.303–3.786) 0.905 (0.153–5.360)	0.916 0.912	95.809
Dominant	AG+GG vs. AA	1.033 (0.305–3.496)	0.959	93.851
Recessive	GG vs. AA+AG	0.865 (0.180–4.150)	0.856	93.820
Overdominant	AG vs. AA+GG	1.108 (0.364–3.373)	0.857	93.821
Additive	G	0.972 (0.421–2.248)	0.948	92.849
*TERC* rs12696304				
Codominant	CG vs. CC GG vs. CC	0.720 (0.200–2.586) 3.643 (0.782–16.961)	0.6140.100	92.529
Dominant	CG+GG vs. CC	1.097 (0.364–3.304)	0.870	93.827
Recessive	GG vs. CC+CG	4.159 (0.963–17.969)	0.056	90.789
Overdominant	CG vs. CC+GG	0.563 (0.168–1.890)	0.352	92.935
Additive	C	1.513 (0.662–3.458)	0.326	92.917

**Table 4 medicina-58-01240-t004:** Distribution of relative leukocyte telomere lengths by genotypes.

Genotype	RLTL		*p*-Value *
	LHON Group Median (IQR)	Control Group Median (IQR)	
*TEP1* rs1760904			
GG	0.725 (-)	0.441 (0.511)	0.738
GA	4.505 (15.405)	0.696 (0.549)	**<0.001**
AA	-	0.512 (0.858)	NA
*TEP1* rs1713418			
AA	3.474 (16.635)	0.506 (0.682)	**0.011**
AG	7.940 (12.964)	0.644 (0.565)	**<0.001**
GG	1.629 (-)	0.695 (0.573)	0.913
*TERC* rs12696304			
CC	3.097 (3.780)	0.456 (0.652)	**0.005**
CG	12.105 (16.187)	0.694 (0.573)	**0.008**
GG	9.764 (-)	0.982 (1.529)	**0.025**

* Mann-Whitney U test was used.

## Data Availability

Data will be provided if a request is made by editors, reviewers, or scientists.

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
