# Peer review of "Relative Leukocyte Telomere Length and Telomerase Complex Regulatory Markers Association with Leber’s Hereditary Optic Neuropathy"

_medicina, 2022, doi:10.3390/medicina58091240_

Round 1

Reviewer 1 Report

I appreciate the opportunity to review the manuscript submitted to the journal of Medicina.

This is a very interesting and important research article emphasizing the relative leukocyte telomere length and telomerase complex regulatory markers association with LHON.

However, I have some concern that needs to be clarified:

1. The authors mention in the discussion the use of idebenone and its impact or lack of impact on RLTL..., although no results are presented, please clarify

2.  It would be significant to discuss the correlation of the obtained results with the parameters of oxidative stress in LHON.

The article "Oxidative Stress Profile in Genetically Confirmed Cases of Leber's Hereditary Optic Neuropathy", J Mol Neurosci. 2021, 71(5):1070-81." may provide some information to the authors.

Author Response

I appreciate the opportunity to review the manuscript submitted to the journal of Medicina.

This is a very interesting and important research article emphasizing the relative leukocyte telomere length and telomerase complex regulatory markers association with LHON.

Dear Reviewer, we appreciate your comments and suggestions. We have made some revision in response to your comments.

However, I have some concern that needs to be clarified:

  1. The authors mention in the discussion the use of idebenone and its impact or lack of impact on RLTL..., although no results are presented, please clarify

In our study RLTL was independent of Idebenone treatment.

  1. It would be significant to discuss the correlation of the obtained results with the parameters of oxidative stress in LHON.

The article "Oxidative Stress Profile in Genetically Confirmed Cases of Leber's Hereditary Optic Neuropathy", J Mol Neurosci. 2021, 71(5):1070-81." may provide some information to the authors.

Thank you, information about oxidative stress was added to the manuscript.

Reviewer 2 Report

This manuscript presents results of the original investigation in the field of genetics of Leber Hereditary Optic Neuropathy (LHON).

The methods used in the research are adequate, the results are presented clearly and critically discussed.

Cited literature is contemporary and from the field. In this case - control study authors compared relative leukocyte telomere length (RLTL) and genotypes of telomerase complex regulatory markers TERC and TEP1 in a group of LHON patients (n = 14) vs. control group (n = 130).

They found significantly longer RLTL in LHON patients, particularly in the older group (more than 32 years of age); RLTL was independent of Idebenone treatment.

In addition, among four analyzed markers, gene polymorphism TERC rs12696304 was the most associated with RLTL.

The obtained results have importance for basic science and for knowledge of LHON ethiopathogenesis, and they require further investigation.

Author Response

Dear Reviewer, thank you very much for your comments.